# Food Fraud in Plant-Based Proteins: Analytical Strategies and Regulatory Perspectives

**DOI:** 10.3390/foods14091548

**Published:** 2025-04-28

**Authors:** Jun-Hyeok Ham, Yeon-Jung Lee, Seung-Su Lee, Hae-Yeong Kim

**Affiliations:** Department of Food Science and Biotechnology, Kyung Hee University, Yongin 17104, Republic of Korea; wns5126@khu.ac.kr (J.-H.H.); oz3473@khu.ac.kr (Y.-J.L.); pastel8981@naver.com (S.-S.L.)

**Keywords:** plant-based proteins, food fraud, food authenticity, adulteration detection

## Abstract

Food fraud and adulteration have been persistent issues affecting food supply chains throughout history. They intensify in parallel with the continuous growth in the global food market. Plant-based proteins, which are recognized as sustainable alternatives, face increased food fraud risks because of disparities in the cost of raw materials and complex processing methods. Despite these challenges, most efforts toward preventing food fraud and developing detection technologies have largely focused on animal-based products, with limited attention given to plant-based proteins. This comprehensive review systematically examines the characteristics of major plant protein sources and explores documented instances of food fraud (e.g., ingredient substitution, adulteration with lower-cost alternatives, and mislabeling) within this sector. Furthermore, we discuss key analytical techniques in detecting food fraud, including chromatography, DNA analysis, spectroscopy, and imaging-based approaches, examining their applications and effectiveness. A systematic literature review was conducted using structured search strategies across Scopus, Web of Science, and PubMed, covering publications from 2010 to 2025 and incorporating keywords related to plant-based proteins, food fraud, adulteration, and authentication, thereby ensuring methodological rigor and comprehensive coverage. This study provides a foundational framework to strengthen food fraud prevention strategies and uphold the integrity of the expanding plant-based protein market.

## 1. Introduction

Alternative foods have recently emerged as a significant research topic of studies addressing global challenges related to population growth and increasing meat consumption [1,2,3]. Conventional animal-derived foods are increasingly being evaluated for their negative environmental, economic, and ethical impacts, as well as their sustainability challenges [4,5]. Consequently, interest in alternative proteins has surged, particularly plant-based proteins, which serve as viable, high-quality substitutes for conventional animal proteins [6,7]. Plant-based proteins encompass a wide variety of sources, including cereals (wheat, rice, oat, and barley), legumes (pea, soybean, lupin, and chickpea), pseudocereals (quinoa and amaranth), and nuts (almonds, walnuts, pistachios, and hazelnuts) [8]. Plant-derived proteins offer significant advantages, such as a lower environmental impact, cost-effectiveness, and consumer acceptance based on perceived health benefits and sustainability factors [6,9].

The development of alternative foods using plant proteins involves several critical processing stages, including raw material selection, extraction, isolation, and various texturizing and flavoring techniques designed to mimic the sensory attributes of animal-derived products [3,10]. However, these complex processing stages, which often involve complicated transformations that alter the original properties of the raw materials, inherently introduce vulnerabilities, which creates opportunities for food fraud [11]. Food fraud, defined as the deliberate misrepresentation or adulteration of food products for economic gain, has a long historical precedent and continues to be a prevalent global issue [12,13]. The economic value of plant-based proteins varies significantly depending on the source plant, with certain ingredients commanding premium prices due to their healthful functional properties [14]. For instance, chickpea flour, which commands a higher economic value due to its favorable nutritional profile, is sometimes adulterated with lower-cost alternatives like grass pea flour, and premium rice varieties such as basmati are occasionally mixed with common rice to artificially increase volume and profitability [15]. Such economic incentives have intensified the risk of fraudulent activities, underscoring the urgent need for systematic approaches to prevent fraud within the rapidly expanding alternative protein market [16].

Analytical approaches employed to detect food fraud, including fraud associated with plant-based proteins, encompass chromatography-based, DNA-based, spectroscopy-based, and imaging-based methods (Figure 1). Chromatography-based and DNA-based techniques are characterized by their high precision and accurate detection capabilities, making them particularly effective in laboratory environments. Conversely, spectroscopy-based and imaging-based methods offer advantages such as non-destructive analysis and high suitability for on-site applications. Each approach has distinct advantages depending on the specific context, and, therefore, these methods have been extensively utilized in detecting food fraud [17]. However, despite their effectiveness, these analytical techniques have primarily been applied to detect fraud in animal-derived food products, and comprehensive studies specifically addressing food fraud related to plant-based protein sources remain limited.

This review first comprehensively examined the characteristics and economic implications of the major plant protein sources used in alternative foods. It then critically reviews documented instances and associated risks of economically motivated adulteration, mislabeling, and allergen contamination within the plant-based protein industry. Furthermore, advanced analytical approaches for detecting plant-based protein fraud, including chromatography-based, DNA-based, spectroscopy-based, and imaging-based methods, are comprehensively evaluated, highlighting their practical applications, strengths, and limitations. Finally, we conclude by discussing prospective insights and future research directions aimed at reinforcing integrity and transparency within the rapidly expanding plant-based protein market. This review provides a foundational analysis for enhancing preventive strategies and ensuring the integrity of the expanding plant-based alternative food sector.

## 2. Methodological Framework for Literature Review

This review employs a systematic literature review (SLR) approach, drawing upon established methodological frameworks [18,19], to comprehensively examine food fraud in plant-based proteins (Figure 2). To structure the investigation, the following two primary research questions were posed: (i) What are the common forms and documented instances of food fraud in plant-based protein products? (ii) What analytical methodologies are utilized to detect adulteration and authenticate plant-derived protein sources? These questions informed both the search strategy and the inclusion/exclusion criteria. Accordingly, we constructed structured Boolean search queries combining the following terms: (“plant-based protein” OR “plant protein” OR “alternative protein” OR “alternative food”) AND (“food fraud” OR “adulteration” OR “authentication”). Searches were performed in Scopus, Web of Science (WoS), and PubMed, which were selected for their comprehensive disciplinary coverage and academic rigor, with the search period spanning from 2010 to 2025.

The identified articles underwent a rigorous multi-step screening and evaluation process. The initial quality verification ensured methodological rigor, scientific validity, and relevance to the review’s scope. Subsequent title screening ensured alignment with the research objectives, and clearly irrelevant studies were excluded. Articles meeting these criteria proceeded to full-text evaluation, where comprehensive assessment confirmed their applicability. Studies failing to meet the inclusion standards at any stage were excluded from the final selection.

In addition to the primary database search, a complementary “snowballing” approach was adopted [20], leveraging reference lists and citations from key articles identified during the primary search. This approach facilitated the identification of additional pertinent publications beyond the original search parameters, including studies published before 2010, provided they offered significant foundational or contextual value. The final selection comprises methodologically sound and contextually relevant publications that collectively enhance understanding and provide substantial insight into food fraud detection and authentication practices within the plant-based protein sector.

## 3. Comprehensive Review of Major Plant Protein Sources

The protein composition and characteristics of major plant-based sources must be established, as variations in these factors and their economic value create risks of food fraud [21]. Plant-based proteins are typically categorized into the following four major classes based on their solubility and extraction properties: albumins (water-soluble), globulins (soluble in dilute salt solutions), prolamins (soluble in aqueous alcohol solutions), and glutelins (soluble or insoluble in dilute acid or alkali solutions) [3,10]. The primary plant protein sources frequently used in alternative foods include legumes, cereals, and nuts (Table 1). Legumes predominantly contain storage proteins, such as globulins and albumins, whereas cereals and nuts mainly contain globulins and albumins [22,23,24]. Development strategies for enhancing the overall nutritional value and functionality of alternative foods often involve combining multiple plant protein sources [25].

Protein extraction from plant sources is a critical step determining the nutritional, functional, and economic value of plant-based protein ingredients. Extraction methods can be broadly classified into dry fractionation and wet extraction processes, each with distinct advantages and limitations [26]. Dry fractionation typically involves mechanical milling and air classification, utilizing differences in particle sizes and densities to enrich proteins without substantial water use, making it highly sustainable. However, this approach generally yields a lower protein purity due to incomplete separation from other components [27]. In contrast, wet extraction methods, particularly alkaline solubilization coupled with isoelectric precipitation, are commonly used to achieve higher purity protein isolates (>95%) [28]. In these processes, proteins are initially solubilized under conditions different than their isoelectric points to remove insoluble impurities and subsequently precipitated by adjusting pH near their isoelectric points, followed by separation and drying. Wet extraction methods often incorporate auxiliary technologies, including enzymatic hydrolysis, ultrasound, microwave, or high-pressure techniques, to enhance cell wall disruption and improve protein recovery rates. These modern technologies significantly increase the extraction efficiency; however, they can incur higher operational costs and complexity. Therefore, the selection of extraction methods depends critically on the target plant source, intended product application, economic considerations, and desired functional properties of the final protein ingredient. These extraction strategies have been comprehensively reviewed in previous studies, offering valuable insight into their optimization across diverse plant protein sources and applications [26,29].

### 3.1. Soybean

Soybean is the most widely cultivated legume globally and serves as an important alternative to animal-derived proteins [30]. Soybean proteins are commonly processed into isolates, concentrates, and flours because of the plant’s excellent functional properties, such as water-holding, emulsification, fat absorption, and gel-forming abilities [30,31,32]. Soy-based foods are nutritionally valuable because of their high contents of essential amino acids, such as lysine and threonine. Their low fat and cholesterol contents also have significant health benefits [33]. Furthermore, soybean is highly cost-effective because of its relatively low price, making it the most widely used raw material in the development and commercialization of alternative foods [34]. Nevertheless, soybean protein is closely associated with allergenic responses, raising safety concerns, as approximately 0.5% of the global population exhibits allergic reactions to soy [35]. Consequently, extensive efforts have been made to explore alternative protein sources and develop processing techniques for reducing allergenicity [36,37].

### 3.2. Pea

Pea is another widely used plant-based protein source that contains essential amino acids, such as lysine, threonine, valine, leucine, isoleucine, and phenylalanine, making it a viable nutritional substitute for animal proteins [22,38,39]. In addition to its nutritional benefits, pea protein is highly valued for its cost-effective, genetically modified organism (GMO)-free, and cholesterol-free qualities, as well as its relatively low allergenic potential when integrated into plant-based food formulations [40]. Pea proteins also exhibit various functional properties, including gel formation, foam stabilization, and emulsification [41]. Processing techniques, such as high-moisture extrusion, are used to modify protein–water interactions in pea-based proteins to achieve fibrous structures similar to meat [7]. These functional characteristics demonstrate the significant potential for creating pea-based alternative foods with superior textural and sensory attributes.

### 3.3. Lupin

Lupin is emerging globally as a significantly promising plant-based protein source because of its excellent sustainability and relatively low production costs compared with other legumes. It offers diverse health benefits, including cholesterol reduction, making it a highly beneficial protein source for human consumption [42]. Lupin has similar protein content characteristics as other legumes, such as limited cysteine and methionine but notably high lysine levels [43]. This distinctive amino acid profile facilitates complementary nutritional interactions, particularly when combined with cereal proteins that are abundant in sulfur-containing amino acids, which enhance the overall dietary balance [44]. However, the usefulness of lupin as a plant protein source is complicated by its significant cross-reactivity with soybean and peanut allergens. Approximately 20–30% of individuals allergic to lupin also exhibit cross-reactivity to these commonly allergenic legumes [45,46]. Despite these issues, the nutritional and economic potential of lupin as an alternative protein source remains significant, highlighting its value in ongoing technological advancements and studies [47].

### 3.4. Chickpea

Chickpea is the third most widely cultivated legume globally after soybean and pea, with significant production in Africa and Asia [48]. Its higher protein bioavailability and lower allergenic potential compared to soybeans enhance its appeal as an alternative protein source [49,50]. Chickpea proteins include globulins (56%), albumins (12%), glutelins (18%), and prolamins (3%) and contain essential amino acids, such as isoleucine, lysine, and tryptophan [48]. In addition to its nutritional profile, chickpea proteins and their derived peptides exhibit bioactive properties associated with health benefits, such as antihypertensive, hypocholesterolemic, and hypolipidemic effects [51,52,53]. These nutritional and functional benefits establish the value of chickpea in the competitive alternative food industry.

### 3.5. Wheat

Wheat is one of the most widely cultivated cereals globally and is an economically accessible and significant source of storage proteins [54]. Depending on the kernel hardness class, wheat varieties are classified into soft, hard, and durum types, with the total protein content ranging from approximately 8% to 18% [41,55]. Wheat proteins contain almost all essential amino acids, except lysine, and exhibit diverse bioactive properties [54]. Approximately 70–80% of wheat proteins consist of gluten-forming proteins, primarily gliadin and glutenin, which can form robust three-dimensional networks through hydrogen bonds, disulfide bonds, and hydrophobic interactions [26,56]. Disulfide bond formation significantly contributes to the structural stability and functionality of wheat gluten, facilitating the effective mimicking of the fibrous texture characteristic of meat [33]. However, wheat gluten poses significant concerns related to allergenicity and dietary sensitivities, notably celiac disease [8]. Many studies have investigated alternative cereal proteins and novel processing techniques to reduce gluten-related health risks and preserve the sensory qualities of wheat-based products [57].

### 3.6. Oat

Oat exhibits considerable commercial potential due to its minimal gluten protein content and the absence of the major allergenic proteins typically found in legumes [58]. The protein content of oat typically ranges from 12% to 20% [59] and has a relatively lower essential amino acid content (~32%) than soybean proteins (~60%), with notably low levels of lysine and sulfur-containing amino acids [60]. Such nutritional constraints necessitate combining oat proteins with other plant protein sources, such as peas, to ensure products with adequate nutritional content. Despite these constraints, oat possesses advantageous functional properties, including high thermal stability, desirable gel formation, and fiber mimicry potential [58]. These characteristics make oats a promising gluten-free alternative, thereby promoting their commercial application in plant-based food formulations.

### 3.7. Rice

Rice is one of the most widely consumed grains globally, and it provides economically accessible and stable plant-based protein ingredients. Unlike other cereal grains, such as wheat, rice is gluten-free, exhibiting a lower risk of allergenicity and better digestibility [61]. Rice protein contains high levels of glutamine (15–31%), proline (12–14%), and leucine (7–14%) and relatively low levels of essential amino acids, such as lysine (1.4–3.3%), tryptophan (0.2–1.0%), and methionine (1.3–2.9%) [62,63]. Thus, to address amino acid imbalances, combining rice protein with legume proteins, such as soybean or pea protein, is a widely adopted complementary strategy in alternative food formulations [64]. Nevertheless, the development of rice proteins is currently affected by technological limitations in food applications because of their poor solubility and other functional inadequacies. Enzymatic hydrolysis, high-pressure processing, and ultrafine grinding have been applied to improve rice protein solubility and expand its food applications [65].

### 3.8. Nuts

Nuts (such as almonds, walnuts, pistachios, and hazelnuts) have become indispensable ingredients in plant-based beverages and alternative foods, owing to their distinctive flavors, creamy textures, and beneficial fatty acid profiles. They are rich in protein and provide a valuable source of essential amino acids, which offer significant benefits for cardiovascular health [66]. Moreover, nuts contain tocopherols, B vitamins, and carotenoids. In particular, almond and hazelnut emulsions are rich in essential minerals, making them suitable milk alternatives for those with celiac disease or lactose intolerance [67,68]. Nut proteins possess high solubility and excellent emulsifying properties, making them effective emulsifiers and foam stabilizers, which enhance their potential use in various food formulations [69]. However, major nut storage proteins, such as 2S albumins, globulins, nonspecific lipid transfer proteins, and oleosins, have potent allergenic properties that limit their broader application [70]. Consequently, structural modifications and enzymatic hydrolysis have been actively explored to reduce allergenicity and improve the safe use of nut proteins in alternative foods [71].foods-14-01548-t001_Table 1Table 1Nutritional and functional properties of major plant proteins.CategoryPlant SourceNutritional PropertiesFunctional PropertiesReferencesLegumesSoybeanHigh in essential amino acids (lysine and threonine); low in fat; rich in isoflavonesStrong emulsification, gelation, and water-holding capacity; commonly used in meat analogues[30,31,32,33,34]PeaRich in lysine, threonine, valine, and leucine; non-GMO, cholesterol-free, and cost-effective protein sourceGel formation, foam stabilization, and emulsification; high-moisture extrusion enables fibrous meat-like textures[7,39,40,41]LupinGood lysine content but low in cysteine and methionineStrong emulsification and gelling properties; enhances nutritional value when combined with cereals[42,43,45,46]ChickpeaRich in isoleucine, lysine, and tryptophan; lower allergenicity compared to soybean; contains bioactive peptidesHypolipidemic and antihypertensive effects; good protein bioavailability[48,49,50,51,52,53]CerealsWheatHigh in gluten-forming proteins (gliadin and glutenin); rich in essential amino acids, except lysineStrong viscoelastic properties due to gluten network formation; ideal for bakery and texturized protein products[33,54]OatModerate protein content (12–20%); low gluten; limited levels of sulfur-containing amino acidsHigh thermal stability; good gel formation[58,59,60]RiceGluten-free; rich in glutamine and proline; low in lysine and methionine; high digestibilityPoor solubility; improved via enzymatic hydrolysis and high-pressure processing[61,62,65]NutsAlmond, walnut, pistachio, hazelnutHigh protein content with balanced amino acid profile; rich in tocopherols, B vitamins, and beneficial fatty acidsStrong emulsifying and foaming properties; used in plant-based beverages and dairy alternatives[67,68,69]

## 4. Food Fraud Risks Associated with the Plant-Based Protein Industry

Food fraud has been conducted throughout history. Although its scale is difficult to quantify, estimates indicate that fraudulent activities impact up to 10% of commercially available food products, with an annual economic burden of USD 10 to 15 billion [72]. Among various forms of food fraud, protein adulteration remains a persistent issue, driven by economic incentives and limitations in conventional protein assessment methods [73]. One of the most infamous cases of protein adulteration is the melamine contamination scandal [74]. Melamine, a nitrogen-rich compound, was used to falsify the protein content measurements in dairy- and plant-based protein products, exploiting the vulnerabilities in the Kjeldahl and Dumas techniques, among others [75,76]. In 2008, melamine-adulterated milk formula in China resulted in the hospitalization of over 300,000 infants, including six fatalities [77]. A similar case occurred in 2007 when melamine-contaminated wheat gluten used in pet food caused acute renal failure in pets across North America [78]. These incidents underscore the critical risks of relying exclusively on indirect protein content assessments and emphasize the urgent need for developing and integrating robust prevention strategies in the alternative protein industry [15].

As the plant-based alternative food market grows and raw materials diversify, new opportunities emerge to meet evolving consumer preferences and demands. However, this ongoing diversification also increases food fraud risks, especially for products processed into powdered, ground, or paste forms, where adulteration is difficult to detect visually or sensorially [79,80]. A common practice in food fraud involves substituting premium materials with lower-value ingredients to increase profit margins. This tactic is frequently observed in products made from cereals, legumes, and nuts [15,81]. For instance, chickpea flour has been deliberately adulterated with lower-cost legume flours, such as grass pea (*Lathyrus sativus* L.) or pea (*Pisum sativum* L.) [82]. Similarly, lupin flour is sometimes used as a substitute for soybean protein, and, if not properly labeled, this substitution can lead to misidentification or unintended consumption [83]. Additionally, high-value products like black gram and split dal are commonly adulterated with wheat, whereas lentils are adulterated with kesari dal, which causes lathyrism [84].

A prominent example in the cereal market involves durum wheat, which commands approximately 20% higher prices than common wheat because of its superior quality. This price difference has led to the fraudulent substitution of durum wheat with common wheat, resulting in economic losses and compromised quality [15]. Similarly, premium rice varieties, such as basmati, which has a distinct aroma and texture, can command market prices of up to four times higher than common rice, creating significant economic incentives for fraudulent substitution [80]. Fraudulent practices are also observed in the nut industry, where high-value almonds and their byproducts are sometimes partially or entirely substituted with lower-cost alternatives, such as peanuts or hazelnuts [85]. Furthermore, pistachio products have been adulterated with cheaper ingredients, such as peas, to artificially inflate profits [86].

Food fraud involving plant-based proteins extends beyond adulteration with cheaper ingredients to include false claims of product authenticity. A notable example is the fraudulent labeling of GMO and non-GMO soybeans, which is driven by price disparities and consumer demand for non-GMO products [87]. The higher market value of non-GMO soy creates economic incentives for fraudulent mislabeling, with instances of GMO soy being falsely marketed as non-GMO to exploit the premium status of non-GMO soy [15,88]. This type of fraud deceives consumers and compromises the transparency of the supply chain.

Beyond economic consequences, food fraud poses significant health risks, particularly regarding allergenic ingredients and gluten adulteration. The undisclosed presence of, or adulteration with, allergenic components of plant-based proteins can trigger severe allergic reactions in sensitive or allergic individuals [89]. Documented cases include the adulteration of chickpea flour with pea flour and the mixing of quinoa flour with soybean, maize, and wheat flours, all of which increase the risk of allergic reactions due to undeclared allergens [82,90,91]. Similarly, gluten fraud, wherein gluten-containing cereals are intentionally or unintentionally introduced into gluten-free products, further exacerbates public health and regulatory challenges [92]. Notably, oat products marketed as gluten-free have been found to contain wheat, rye, or barley, which are often added to increase product volume or profitability [15]. Such fraudulent practices pose significant risks to individuals with gluten-related disorders.

In conclusion, food fraud within the plant-based protein sector poses multifaceted risks, including economic loss, consumer trust erosion, and significant health hazards (Table 2). Current regulations have primarily focused on basic labeling standards and allergen declarations; however, comprehensive guidelines that explicitly address food fraud prevention in alternative protein products remain insufficient. Governments and international regulatory bodies must effectively address these issues by establishing robust standards and guidelines for the labeling and compositional requirements of plant-based proteins. Furthermore, the implementation of comprehensive traceability systems, from raw material sourcing to finished product distribution, would significantly enhance the verification of the authenticity and safety of plant-based proteins [93]. Accurate detection at the point of occurrence and proactive monitoring immediately prior to consumer distribution are crucial to effectively combat food fraud. These measures would impose significant deterrents on fraudulent actors by increasing the likelihood of detection and execution of penalties, simultaneously safeguarding consumers by ensuring the authenticity of the products they purchase. Thus, the integration of precise analytical detection methods into the supply chain is essential for protecting consumer health and upholding integrity within the plant-based protein industry.
foods-14-01548-t002_Table 2Table 2Categories and characteristics of food fraud in the plant-based protein industry.CategoryCharacteristicsExamplesReferencesEconomically motivated adulteration(EMA)Deliberate substitution, dilution, or misrepresentation of plant protein sources for economic gain.This includes the addition of lower-cost ingredients or inflation of protein content using non-protein nitrogen compounds.Adulteration of high-value plant proteins (e.g., chickpea flour) with lower-cost alternatives (e.g., pea and grass pea).Fraudulent substitution of premium durum wheat with common wheat.Melamine contamination to falsely increase protein content in plant-based protein powders.[77,78,79,82,83,84,85,86]Mislabeling and supply chain integrity violationsFalse claims regarding the origin, processing method, or composition of plant-based protein products. This includes the misrepresentation of genetically modified status, organic labeling fraud, and false species declaration.Fraudulent labeling of GMO soy as non-GMO to exploit premium pricing.False country-of-origin claims, such as selling common rice as premium basmati rice to increase market value.[15,80,87,88]Health and safety risksUnintentional or intentional contamination of plant-based protein sources with allergens, gluten, or toxic compounds, posing risks to consumers.Adulteration of gluten-free oat products with wheat, rye, or barley.Cross-contamination of quinoa flour with undeclared soy, maize, or wheat proteins.Undisclosed presence of allergens in plant-based protein formulations, leading to severe allergic reactions.[89,90,91,92]GMO, genetically modified organism.

## 5. Detection Methods for Food Fraud in Plant-Based Proteins

Adulteration of plant-based proteins is detected using a range of analytical methods, generally classified into chromatography-based, DNA-based, spectroscopy-based, and imaging-based detection techniques. Recent technological advancements have significantly enhanced the practical applicability of analytical techniques for detecting food fraud in plant-based proteins (Table 3). Chromatography-based methods effectively detect complex chemical adulterants and species-specific metabolites. DNA-based techniques have proven crucial for authenticating species and identifying GMOs. Spectroscopy methods are invaluable for rapid screening and non-destructive authentication. Imaging methods have shown promise for real-time detection and high-throughput screening in various product forms. The integration of these complementary approaches will enable comprehensive, robust, and reliable authentication in plant-based protein products, supporting integrity and transparency within the alternative protein market.

### 5.1. Chromatography-Based Detection

Chromatographic techniques coupled with mass spectrometry (MS) are considered the gold standard for identifying food adulterants. These methods are broadly classified into liquid chromatography (LC) and gas chromatography (GC) based on the phase characteristics of their mobile and stationary phases. Chromatographic methods coupled with mass spectrometry demonstrate strengths in high sensitivity, specificity, and reproducibility, making them indispensable tools for food fraud detection. However, matrix effects, sample preparation procedures, and derivatization strategies significantly influence analytical accuracy and precision. Although beyond the scope of this overview, these methodological considerations have been comprehensively discussed in previous studies [94]. Spörl et al. [95] developed a rapid LC-MS/MS multi-method for detecting 23 plant-based protein sources (legumes, oilseeds, and grains) in meat products at high sensitivity (limit of detection [LOD] <200 µg/g). Seki et al. [96] developed an LC–MS/MS-based method capable of simultaneously detecting and quantifying trace contaminations or admixtures of allergenic grains, including wheat, spelt, rye, barley, oats, and buckwheat, in processed foods, with detection limits as low as 0.028–0.056 μg/L. Ning et al. [97] recently established an ultra-performance LC-MS/MS (UPLC-MS/MS) method using specific signature peptides to detect adulteration in plant protein beverages, which has successfully identified almond, peanut, walnut, and soybean adulterants with a limit of quantification of 0.01–0.5 g/L and recoveries of 84.77–110.44%. Russo et al. [98] developed a multiple reaction monitoring-based UPLC-MS/MS method targeting a puroindoline-a-derived peptide, enabling sensitive detection of common wheat adulteration in durum wheat down to 0.01% (100 µg/g), including in processed products like pasta. Farag et al. [99] employed UPLC-MS with chemometric analysis to distinguish lupin and lentil seeds, identifying 66 distinct metabolites that enable reliable authentication and fraud detection in legume ingredients. Additionally, LC-MS/MS has proven effective in detecting adulteration in grain products and identifying nitrogen-rich adulterants commonly used to artificially inflate protein content in foods [100,101,102].

GC-MS primarily profiles volatile organic compounds (VOCs), making it less suitable for direct protein analysis; instead, it is more suitable for the indirect detection of adulteration using characteristic VOC profiles. Shannon et al. [103] developed a two-tiered GC-MS approach using headspace solid-phase microextraction to identify aldehydes and organic acids that distinguish adulterants in authentic basmati rice. Pastor et al. [104] employed GC-MS combined with chemometric analysis to effectively discriminate between cereal (corn, wheat, barley, and oat) and oilseed plant species, enhancing methods for the authentication of plant-based food ingredients. These chromatography-based methods are essential analytical tools in plant-based protein authentication with high sensitivity and specificity, which are crucial for detecting and preventing complex food fraud.

### 5.2. DNA-Based Detection

DNA-based methods offer significant advantages in detecting food fraud due to their high sensitivity, specificity, and ability to identify adulteration even in processed food ingredients. Among these methods, polymerase chain reaction (PCR) is the most commonly employed method, as it amplifies species-specific DNA barcodes, effectively identifying species substitution or adulteration in plant-based proteins. For example, Shin et al. [105] developed a multiplex PCR protocol using two species-specific primers and a plant universal control that demonstrated high specificity without cross-amplification among 22 plants and a detection sensitivity of up to 0.1% adulteration of wheat or soy in pea flour. Real-time quantitative PCR (qPCR) enhances PCR specificity by integrating fluorescence-based detection, allowing for the quantification of adulterant DNA levels. Zheng et al. [106] developed TaqMan qPCR assays targeting unique gene markers of chickpeas, quinoa, and other grains, achieving detection thresholds below 0.01%. Cottenet and Blancpain [107] developed a real-time PCR assay targeting the mitochondrial 16S rRNA gene from vertebrates, effectively detecting vertebrate material contamination in plant-based products at a threshold of 0.1% (*w*/*w*). Carloni et al. [108] developed a multiplex real-time PCR assay that reliably detected and quantified common wheat adulteration in durum wheat-based products at levels as low as 0.15% (*w*/*w* genomic DNA), enabling species-specific DNA assessment in raw materials and flours. Digital PCR (dPCR), especially droplet digital PCR (ddPCR), has emerged as a promising technique due to its ability to provide absolute quantification and exceptional sensitivity. By partitioning samples into thousands of micro-reactions, this method directly counts DNA targets and is, thus, highly effective in verifying non-GMO plant protein authenticity, for which precise quantification is key for regulatory compliance. Long et al. [109] developed a duplex ddPCR method that accurately quantified five genetically modified soybean events, demonstrating higher sensitivity and throughput than conventional PCR techniques, thereby aligning with stringent international GMO labeling requirements. Furthermore, Demeke and Eng [110] successfully employed two multiplex ddPCR assays to simultaneously detect and quantify 19 soybean GMO events, which demonstrated efficiency and high sensitivity in detecting GMO adulterants at thresholds as low as 0.01%. Beyond GMO detection, ddPCR also allows for the precise identification and quantification of allergenic grains and their contamination in grain-based products. Schulze et al. [111] developed a validated ddPCR assay that accurately quantified common wheat, durum wheat, rye, and barley, which are all recognized as allergenic cereals, and reliably detected their presence at levels as low as 1% in adulterated durum wheat.

In addition to PCR, DNA barcoding and next-generation sequencing (NGS) have been employed to simultaneously authenticate multiple species. DNA barcoding sequences standardized gene regions (such as rbcL or ITS) and compare them against reference databases to effectively detect species admixture or substitution. Amane and Ananthanarayan [84] successfully employed the DNA barcoding loci rbcL and trnH-psbA to detect refined wheat flour and white pea flour adulteration in black-gram-based products at 5% contamination levels. NGS metabarcoding can simultaneously read DNA from all species found in a product to reveal complex adulteration. Faller et al. [112] used NGS to determine adulteration in plant-based protein powder supplements, uncovering minor species contaminations below 1% of the total DNA content, demonstrating the method’s sensitivity and comprehensiveness in revealing potential adulterants. Collectively, DNA-based methods are essential safeguards against food fraud, ensuring the authenticity and integrity of plant-based protein products in an increasingly complex market.

### 5.3. Spectroscopy-Based Detection

Spectroscopy-based techniques, notably Fourier-transform infrared (FT-IR) and near-infrared (NIR), have emerged as rapid, non-destructive tools for verifying authenticity and detecting adulteration in plant protein products. These methods capture unique chemical fingerprints for both targeted identification of known adulterants and untargeted screening for potential anomalies. FT-IR spectroscopy measures the absorption of infrared light through molecular bonds, generating characteristic spectra that serve as molecular fingerprints of plant protein samples. This makes FT-IR highly suitable for rapid, non-destructive, and cost-effective authentication. Rodríguez et al. [90] successfully used FT-mid-IR (FT-MIR) spectroscopy combined with chemometric classification models to effectively distinguish pure quinoa flour from samples adulterated with soybean, maize, or wheat flour at levels of 1–10% *w*/*w*, achieving classification error rates between 2 and 8%. Arslan et al. [113] used FT-IR to detect the adulteration of wheat flour with barley flour, achieving precise quantification of barley adulterants down to approximately 0.30%.

NIR, which operates within the 700–2500 nm wavelength, probes overtone vibrations of CH, NH, and OH bonds and excels in rapid, bulk analysis with minimal preparation. It can penetrate deeper into samples than MIR spectroscopy. Dayananda et al. [114] employed NIR spectroscopy to effectively discriminate binary mixtures of chickpea, corn, and tapioca flours, demonstrating its potential as an initial screening method for plant protein authenticity verification. Bala et al. [115] developed an NIR-based model and successfully quantified maize flour adulteration of chickpea flour within 1–90% *w*/*w*, achieving a coefficient of determination of 0.999. López et al. [116] demonstrated that NIR spectroscopy combined with Soft independent modeling of class analogy (SIMCA) effectively classified hazelnut samples adulterated with almond or chickpea at 3% levels, showing over 94% accuracy in detecting adulterated samples. Miaw et al. [117] developed a portable NIR spectroscopy method that accurately detected cashew adulteration with allergenic nuts (peanut, Brazil nut, macadamia, pecan) at 0.1–10% *w*/*w*, enabling rapid, non-destructive allergen screening.

Researchers often augment NIR spectroscopy applications with Fourier-transform methods (FT-NIR). Aykas and Menevseoglu [118] applied portable FT-MIR and FT-NIR spectroscopy with chemometrics to detect green pea and peanut adulteration in pistachio powder at concentrations of 5–40% *w*/*w*, thereby achieving high predictive accuracy (rVal > 0.99). Furthermore, De Géa Neves et al. [119] utilized FT-NIR spectroscopy combined with chemometric techniques to authenticate plant-based protein powders and accurately classify adulterants, such as whey, soy, and wheat, at adulteration levels of 10–40% and 100% sensitivity and specificity. These spectroscopy-based techniques offer promising solutions through their high accuracy, sensitivity, and portability, which enables flexible and on-site verification of food authenticity and integrity.
foods-14-01548-t003_Table 3Table 3Analytical methods for detecting food fraud in plant-based proteins.Analytical MethodsFoodAdulterant and FraudReferencesChromatography-basedLC-MS/MSMeat productsAdulteration with 23 plant-based proteins (LOD < 200 µg/g)[95]LC-MS/MSPlant-based productsDetection of grains (buckwheat, wheat, rye, barley, oats); LOD between 0.028 and 0.056 g/L[96]LC-MS/MSGrain productsNitrogen-rich adulterants to artificially inflate protein content[100]UPLC-MSLupin and lentil seedsEight types of lupin and lentil seeds[99]UPLC-MS/MSPlant protein beveragesAlmond, peanut, walnut, and soybean adulteration (LOQs between 0.01 and 0.5 g/L)[97]UPLC-MS/MSDurum wheatCommon wheat(LOD < 100 µg/g)[98]GC-MSBasmati riceSeven different rice varieties[103]GC-MSCereal grains and oilseed plantsDifferentiation among plant species[104]DNA-basedPCRPea flourWheat or soy adulteration (LOD 0.1%)[105]qPCRChickpea, quinoa, coix seed, and rice Detection of unique gene markers (LOD < 0.01%)[106]qPCRPlant-based productsVertebrate contamination (threshold 0.1%)[107]qPCRDurum wheatCommon wheat(LOD < 0.15%)[108]ddPCRSoybeanQuantification of GMO adulterants (LOD 0.01%)[110]ddPCRSoybeanQuantification of five genetically modified soybean events (LOQ 0.1%)[109]ddPCRDurum wheatCommon wheat, rye, and barley(LOD < 1%)[111]DNA BarcodingBlack-gram-based productsDetection of wheat and white pea flour adulteration (5% contamination)[84]NGSPlant protein powder supplementsDiverse species contamination (soybean, chia seeds, quinoa, etc.)[112]Spectroscopy-basedFT-IRWheat flourAdulteration with barley flour (0.30% detection)[113]FT-MIRQuinoa flourAdulteration with soybean, maize, and wheat flours (1–10%)[90]NIRChickpea flourMaize flour adulteration (1–90%)[115]NIRHazelnutAlmond or chickpea adulteration (3%)[116]NIRCashewAdulteration with peanut, Brazil nut, macadamia, and pecan (0.1–10%)[117]FT-NIRPistachio powderAdulteration with green pea and peanut (5–40%)[118]FT-NIRPlant protein powdersAuthentication of whey, soy, and wheat adulteration (10–40%)[119]Imaging-basedHSIQuinoa flourDetection of wheat, rice, soybean, and corn contamination with chemometric analysis (R^2^ = 0.99)[120]HSIWheat flourPeanut and walnut adulteration detection (LOD 0.03%)[121]SWIR-HSIAlmond powderPeanut adulteration detection (100% specificity)[122]VNIR-HSIGround beefSoy protein adulteration (LOD 0.74%)[123]HSIWheat flourPeanut and walnut powder adulteration detection (LOD 0.5%)[124]Visible Imaging with AIRice varietiesAuthentication and fraud detection (93–99% accuracy)[125]Multiple Imaging SensorSkimmed milk powderDetection of plant protein adulterants (10–50%)[126]LC-MS/MS, liquid chromatography–tandem mass spectrometry; GC-MS, gas chromatography–mass spectrometry; UPLC-MS/MS, ultra-performance liquid chromatography–tandem mass spectrometry; PCR, polymerase chain reaction; qPCR, quantitative polymerase chain reaction; ddPCR, droplet digital polymerase chain reaction; DNA, deoxyribonucleic acid; NGS, next-generation sequencing; FT-IR, Fourier-transform infrared spectroscopy; FT-MIR, Fourier-transform mid-infrared spectroscopy; NIR, near-infrared spectroscopy; FT-NIR, Fourier-transform near-infrared spectroscopy; HSI, hyperspectral imaging; SWIR-HSI, short-wave infrared hyperspectral imaging; VNIR-HSI, visible–near-infrared hyperspectral imaging; AI, artificial intelligence; GMO, genetically modified organism; LOQ, limit of quantification; LOD, limit of detection; R^2^, coefficient of determination.

### 5.4. Imaging-Based Detection

Imaging techniques have become indispensable for detection of food fraud, offering non-destructive and rapid analyses by combining visual and chemical information. Typically, these imaging methods are integrated with artificial intelligence (AI) algorithms, which further enhance accuracy, speed, and reliability in fraud detection. Hyperspectral imaging (HSI) is a key technique in food fraud detection, simultaneously capturing spectral and spatial data to distinguish authentic ingredients from substitutes. Wu et al. [120] successfully utilized portable HSI combined with partial least squares regression (PLSR) to detect adulteration in quinoa flour, demonstrating high prediction accuracy (R^2^ = 0.99). Zheng et al. [121] demonstrated that HSI accurately detected and classified peanut and walnut adulterants in wheat flour at concentrations as low as 0.03% (*w*/*w*), highlighting its utility for non-destructive allergen screening in grain-based products. Faqeerzada et al. [122] applied shortwave IR HSI and a data-driven soft independent modeling of class analogy method for detecting peanut adulterants in almond powder, achieving 100% sensitivity and 89–100% specificity. Jiang et al. [123] combined visible and NIR hyperspectral imaging with PLSR to quantitatively determine soybean protein powder adulteration in ground beef, reporting excellent predictive accuracy (Rp = 0.9933) with an LOD of 0.74% (*w*/*w*). Zhao et al. [124] demonstrated that HSI accurately quantified peanut and walnut powder contamination in whole wheat flour at levels as low as 0.5% (*w*/*w*), providing a non-destructive solution for visualizing allergenic nut contamination in grain-based products.

Conventional visible imaging, combined with advanced image-processing algorithms and AI techniques, is also emerging as a valuable tool for fraud detection. Izquierdo et al. [125] combined conventional visible imaging with deep learning algorithms to authenticate and distinguish five different rice varieties, achieving high accuracy (93–99%). Müller-Maatsch et al. [126] developed a portable, hyphenated optical multi-sensor system that integrated ultraviolet–visible, fluorescence, and NIR spectroscopy with a one-class classification approach to effectively identify fraudulent additions of plant protein powders and nitrogen-rich compounds in skimmed milk powder within an adulteration range of 0.1–50% (*w*/*w*). These imaging and AI-based detection approaches provide powerful complementary techniques that can accurately and noninvasively identify adulteration in plant-based protein ingredients.

## 6. Conclusions and Future Prospects

Historically, analytical methods for food authenticity have been predominantly applied to animal-derived foods, resulting in the limited development of techniques specifically for detecting plant-based protein fraud [127]. However, with the growing consumption and utilization of plant-based proteins, there is an urgent need for advanced analytical strategies specifically tailored to detect fraud within this growing sector. Each analytical technique in Table 4 presents unique strengths and limitations that determine its applicability in plant-based protein fraud detection. Chromatography-based techniques are highly precise and reproducible but require sophisticated laboratory settings, significant costs, and specialized personnel, limiting their practical application in field settings. Similarly, DNA-based approaches demonstrate remarkable sensitivity and applicability in authenticating processed foods but are not always reliable in directly determining protein adulteration and often entail complex laboratory-bound procedures. Conversely, spectroscopy and imaging methods provide rapid, portable, and on-site applicability but typically offer lower accuracy and reproducibility than laboratory-based analyses.

To overcome these limitations, recent research has increasingly investigated the integration of AI technologies into analytical strategies to enhance food fraud detection accuracy and reliability. AI-driven algorithms can easily process large-scale data, enhancing the analytical precision in imaging and spectroscopy to levels comparable to those of traditional laboratory methods and simplifying complex analytical processes in chromatography and genetic detection, which previously required expert knowledge. Therefore, future studies should focus on leveraging AI to complement the strengths of chromatography, genetic detection, spectroscopy, and imaging, mitigating their individual limitations. Through this integrated approach, detection methods can become more rapid and precise, which would ultimately strengthen transparency and integrity in the plant-based protein market.

## Figures and Tables

**Figure 1 foods-14-01548-f001:**
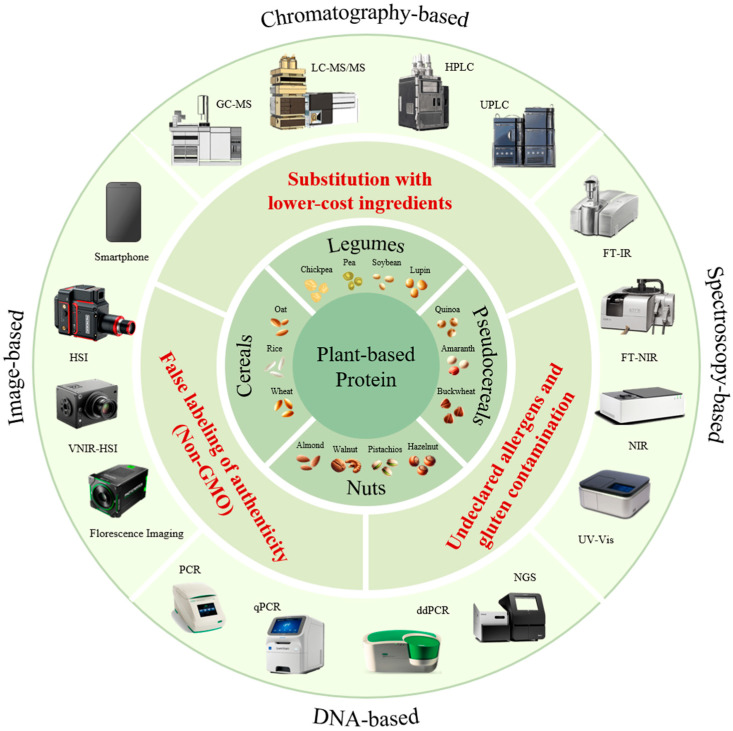
Analytical approaches for detecting food fraud in plant-based proteins.

**Figure 2 foods-14-01548-f002:**
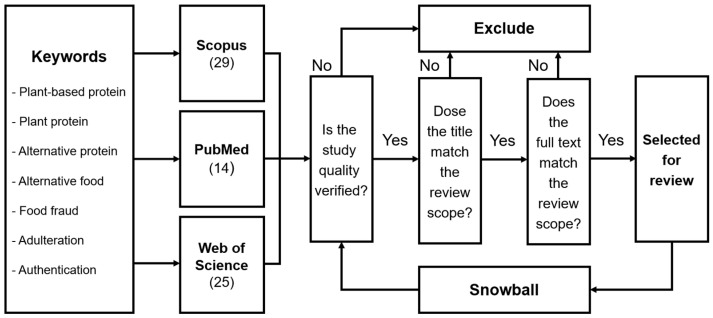
The systematic literature review’s workflow for identifying and selecting relevant studies.

**Table 4 foods-14-01548-t004:** Comparison of analytical techniques for food fraud detection in plant-based proteins.

Category	Chromatography-Based	DNA-Based	Spectroscopy-Based	Imaging-Based
Methods	LC-MS/MS, GC-MS, HPLC, UPLC-MS/MS	PCR, qPCR, ddPCR, NGS	FT-IR, NIR, Raman Spectroscopy	Hyperspectral Imaging, AI-assisted Visible Imaging
Accuracy	High to very high	Very high	Moderate to high	Moderate to high
Analysis speed	Moderate (30 min–2 h)	Fast to slow (30 min–48 h)	Very fast (few seconds–minutes)	Fast (real-time processing)
Cost	High	Moderate to high	Low to moderate	Moderate to high
Major applications	Detection of protein adulteration, nitrogen-rich fraud, species authentication	Species authentication, GMO detection, allergen identification	Rapid screening for ingredient substitution, non-destructive authentication	Real-time food fraud detection, authentication of powdered products
Advantages	High sensitivity and specificity, capable of detecting small molecular adulterants	Highly specific, capable of detecting gene even in processed foods	Non-destructive, rapid analysis, portable instruments available	Provides spatial and spectral data, enables AI-assisted real-time detection
Limitations	High cost, requires specialized training and equipment	Cannot directly detect protein adulteration, requires intact DNA	Limited sensitivity for minor adulterants, may require extensive calibration	High cost, requires AI-driven analysis, lower sensitivity for chemical adulteration

LC-MS/MS, liquid chromatography–tandem mass spectrometry; GC-MS, gas chromatography–mass spectrometry; HPLC, high-performance liquid chromatography; UPLC-MS/MS, ultra-performance liquid chromatography–tandem mass spectrometry; PCR, polymerase chain reaction; qPCR, quantitative polymerase chain reaction; ddPCR, droplet digital polymerase chain reaction; NGS, next-generation sequencing; FT-IR, Fourier-transform infrared spectroscopy; NIR, near-infrared spectroscopy; AI, artificial intelligence; GMO, genetically modified organism.

## Data Availability

No new data were created or analyzed in this study. Data sharing is not applicable to this article.

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
