# Peer review of "Food Fraud in Plant-Based Proteins: Analytical Strategies and Regulatory Perspectives"

_foods, 2025, doi:10.3390/foods14091548_

Round 1
Reviewer 1 Report
Comments and Suggestions for Authors
Journal: Foods
Manuscript ID: foods-3602267
Title: Food Fraud in Plant-Based Proteins: Analytical Strategies and Regulatory Perspectives
Dear Author,
The manuscript entitled “Food Fraud in Plant-Based Proteins: Analytical Strategies and Regulatory Perspectives” describes a foundational framework to strengthen food fraud prevention strategies and uphold the integrity of the expanding plant-based protein market. The manuscript is engaging; however, a few sections could benefit from improvement. As a result, a minor revision is recommended. Please refer to the comments below for further details:
- Line 34: please summarize figure 1 with additional information. Write analytical approaches for detecting food fraud in plant-based proteins. Also write food fraud in plant-based proteins.
- In the introduction section, a summary and explanation of the topics covered in the subsequent subheadings are provided. Please include this information for clarity.
- Line 74-75: title of table should be above.
- Line 270: Please summarize and interpret the information presented in Tables 3 and 4.
- Lines 274, 300, 340, and 373 (Sections 4.1, 4.2, 4.3, and 4.4) require additional supporting literature or further illustrative examples to strengthen the discussion.
- The references cited are relevant and appropriately support the content of the manuscript.
Author Response
The manuscript entitled “Food Fraud in Plant-Based Proteins: Analytical Strategies and Regulatory Perspectives” describes a foundational framework to strengthen food fraud prevention strategies and uphold the integrity of the expanding plant-based protein market. The manuscript is engaging; however, a few sections could benefit from improvement. As a result, a minor revision is recommended. Please refer to the comments below for further details:
Comment 1) Please summarize figure 1 with additional information. Write analytical approaches for detecting food fraud in plant-based proteins. Also write food fraud in plant-based proteins. In the introduction section, a summary and explanation of the topics covered in the subsequent subheadings are provided. Please include this information for clarity.
Response to Comment 1) We sincerely thank the reviewer for this insightful comment and valuable suggestion. In response, we have revised the Introduction section to provide a more specific and comprehensive explanation of Figure 1, as well as to clearly present the analytical approaches commonly used to detect food fraud in plant-based proteins.
Specifically, we have added the following description (Lines 58–69) to contextualize the four major analytical categories: chromatography-based, DNA-based, spectroscopy-based, and imaging-based techniques. We also noted that, although these methods are widely applied in food fraud detection, they have been comparatively underutilized in the context of detecting adulteration and misrepresentation in plant-based proteins, in order to underscore the need for increased attention and tailored methodological development in this area.
Additionally, to address the reviewer’s comment, “Also write food fraud in plant-based proteins,” we have incorporated a representative case of plant protein adulteration earlier in the paragraph (Lines 51–54). These illustrative examples highlight the typical patterns of food fraud in plant-based proteins, while a broader range of documented incidents and associated risk types are comprehensively discussed in Section 4 (Food Fraud Risks Associated with the Plant-Based Protein Industry).
Lastly, to enhance clarity and to better reflect the organization of the review, we revised the final paragraph of the Introduction to explicitly outline the content of each major section (Lines 70–81). We greatly appreciate the reviewer’s thoughtful feedback, which helped improve the clarity, structure, and scientific rigor of the manuscript.
Lines 58-69) Analytical approaches employed in detecting food fraud, including fraud associated with plant-based proteins, encompass chromatography-based, DNA-based, spectroscopy-based, and imaging-based methods (Figure 1). Chromatography-based and DNA-based techniques are characterized by their high precision and accurate detection capabilities, making them particularly effective in laboratory environments. Conversely, spectroscopy-based and imaging-based methods offer advantages such as non-destructive analysis and high suitability for on-site applications. Each approach has distinct advantages depending on the specific context, and therefore, these methods have been extensively utilized in detecting food fraud (Vinothkanna et al., 2024). However, despite their effectiveness, these analytical techniques have primarily been applied to detecting fraud in animal-derived food products, and comprehensive studies specifically addressing food fraud related to plant-based protein sources remain limited.
Lines 51-54) For instance, chickpea flour, which commands a higher economic value due to its favorable nutritional profile, is sometimes adulterated with lower-cost alternatives like grass pea flour, and premium rice varieties such as basmati are occasionally mixed with common rice to artificially increase volume and profitability (Faller et al., 2021).
Lines 70-81) This review first comprehensively examined the characteristics and economic implications of the major plant protein sources used in alternative foods. It then critically re-views documented instances and associated risks of economically motivated adulteration, mislabeling, and allergen contamination within the plant-based protein industry. Furthermore, advanced analytical approaches for detecting plant-based protein fraud, including chromatography-based, DNA-based, spectroscopy-based, and imaging-based methods, are comprehensively evaluated, highlighting their practical applications, strengths, and limitations. Finally, we conclude by discussing prospective insights and future research directions aimed at reinforcing integrity and transparency within the rap-idly expanding plant-based protein market. This review provides a foundational analysis for enhancing preventive strategies and ensuring the integrity of the expanding plant-based alternative food sector.
References:
- Faller, A.C.; Kesanakurti, P.; Arunachalam, T. Chapter 14 - Fraud in grains and cereals. In Food Fraud, Hellberg, R.S., Everstine, K., Sklare, S.A., Eds.; Academic Press. 2021, pp. 281-308, https://doi.org/10.1016/B978-0-12-817242-1.00007-5.
- Vinothkanna, A.; Dar, O.I.; Liu, Z.; Jia, A.-Q. Advanced detection tools in food fraud: A systematic review for holistic and rational detection method based on research and patents. Food Chem. 2024, 446, 138893, https://doi.org/10.1016/j.foodchem.2024.138893.
Comment 2) Line 74-75: title of table should be above.
Response to Comment 2) We thank the reviewer for the valuable comment. Following the suggestion, we have relocated the title “Table 1. Nutritional and functional properties of major plant proteins.” above the table, as indicated in Line 147.
Comment 3) Please summarize and interpret the information presented in Tables 3 and 4.
Response to Comment 3) We appreciate the reviewer’s insightful comment regarding the insufficient summary and interpretation of Tables 3 and 4 in the original manuscript. In response, we have revised the manuscript to more clearly integrate both tables into the overall narrative and provide explicit contextual interpretation.
To improve clarity and narrative flow, we have reordered the tables—placing the methodological overview (formerly Table 4) before the comparative summary of analytical features (formerly Table 3). This adjustment allows us to first present specific applications and case-based evidence (now Table 3), followed by a structured synthesis of methodological advantages and limitations (now Table 4), thereby enhancing reader comprehension.
We have substantially revised the main discussion section (Lines 338-347) to provide a clearer interpretation of the revised Table 3, including technological advances and representative examples illustrating each method’s practical application in plant-based protein fraud detection.
Furthermore, we expanded the description of the revised Table 4 within the conclusion section (Lines 521-530), highlighting the practical limitations and field applicability of each technique. By integrating this content into the concluding remarks, we aimed to strengthen the manuscript’s coherence and provide a more holistic view of analytical strategies for fraud detection.
We believe these revisions directly address the reviewer’s concern and improve the manuscript’s clarity, analytical depth, and overall scholarly quality.
Lines 338-347) Recent technological advancements have significantly enhanced the practical applicability of analytical techniques for detecting food fraud in plant-based proteins (Table 3). Chromatography-based methods effectively detect complex chemical adulterants and species-specific metabolites. DNA-based techniques have proven crucial for authenticating species and identifying GMO. Spectroscopy methods are invaluable for rapid screening and non-destructive authentication. Imaging methods have shown promise for real-time detection and high-throughput screening in various product forms. The integration of these complementary approaches will enable comprehensive, robust, and reliable authentication in plant-based protein products, supporting integrity and transparency within the alternative protein market.
Lines 521-530) Each analytical technique in Table 4 presents unique strengths and limitations that determine its applicability in plant-based protein fraud detection. Chromatography-based techniques are highly precise and reproducible but require sophisticated laboratory settings, significant costs, and specialized personnel, limiting their practical application in field settings. Similarly, DNA-based approaches demonstrate remarkable sensitivity and applicability in authenticating processed foods but are not always reliable in directly determining protein adulteration and often entail complex laboratory-bound procedures. Conversely, spectroscopy and imaging methods provide rapid, portable, and on-site applicability but typically offer lower accuracy and reproducibility than laboratory-based analyses.
Comment 4) Lines 274, 300, 340, and 373 (Sections 4.1, 4.2, 4.3, and 4.4) require additional supporting literature or further illustrative examples to strengthen the discussion. The references cited are relevant and appropriately support the content of the manuscript.
Response to Comment 4) We sincerely thank the reviewer for this valuable comment. We agree that further supporting literature can help reinforce the key arguments in Sections 5.1 to 5.4. In response, we have added relevant references that illustrate the practical application of each detection technique for identifying food fraud involving plant-based proteins (Lines 372-376, 380-383, 413-416, 378–381, 429-432, 493-495, and 501-504).
In addition, recognizing that adulteration may result in the unintended presence of allergenic grains, we have expanded the discussion to include studies on the detection of allergenic grain admixture in plant-based protein products. This allows us to address not only economic fraud but also the potential health risks associated with undeclared allergens, thereby complementing the cases discussed in Section 4. We appreciate the reviewer’s suggestion, which has helped us improve the depth and clarity of our discussion.
Lines 372-376) Seki et al. developed an LC–MS/MS-based method capable of simultaneously detecting and quantifying trace contamination or admixture of allergenic grains, including wheat, spelt, rye, barley, oats, and buckwheat, in processed foods, with detection limits as low as 0.028–0.056 μg/L.
Lines 380-383) Russo et al. developed a multiple reaction monitoring-based UPLC-MS/MS method targeting a puroindoline a–derived peptide, enabling sensitive detection of common wheat adulteration in durum wheat down to 0.01% (100 µg/g), including in processed products like pasta.
Lines 413-416) Carloni et al. developed a multiplex real-time PCR assay that reliably detected and quantified common wheat adulteration in durum wheat-based products at levels as low as 0.15% (w/w genomic DNA), enabling species-specific DNA assessment in raw materials and flours.
Lines 429-432) Schulze et al. developed a validated ddPCR assay that accurately quantified common wheat, durum wheat, rye, and barley, which are all recognized as allergenic cereals, and reliably detected their presence at levels as low as 1% in adulterated durum wheat.
Lines 467-473) López et al. demonstrated that NIR spectroscopy combined with Soft independent modelling of class analogy (SIMCA) effectively classified hazelnut samples adulterated with almond or chickpea at 3% levels, showing over 94% accuracy in detecting adulterated samples. Miaw et al. developed a portable NIR spectroscopy method that accurately detected cashew adulteration with allergenic nuts (peanut, Brazil nut, macadamia, pecan) at 0.1–10% w/w, enabling rapid, non-destructive allergen screening.
Lines 493-495) Zheng et al. demonstrated that HSI accurately detected and classified peanut and walnut adulterants in wheat flour at concentrations as low as 0.03% (w/w), highlighting its utility for non-destructive allergen screening in grain-based products.
Lines 501-504) Zhao et al. demonstrated that HSI accurately quantified peanut and walnut powder contamination in whole wheat flour at levels as low as 0.5% (w/w), providing a nondestructive solution for visualizing allergenic nut contamination in grain-based products.
References:
- Seki, Y.; Nakamura, K.; Arimoto, C.; Kikuchi, H.; Yamakawa, H.; Nagai, H.; Ito, T.; Akiyama, H. Development of a simple and reliable high-performance liquid chromatography–tandem mass spectrometry approach to simultaneously detect grains specified in food allergen labeling regulation on processed food commodities. J Chromatogr A. 2021, 1639, 461877, https://doi.org/10.1016/j.chroma.2021.461877.
- Russo, R.; Cusano, E.; Perissi, A.; Ferron, F.; Severino, V.; Parente, A.; Chambery, A. Ultra-high performance liquid chromatography tandem mass spectrometry for the detection of durum wheat contamination or adulteration. J Mass Spectrom. 2014, 49, 1239-1246, https://doi.org/10.1002/jms.3451.
- Carloni, E.; Amagliani, G.; Omiccioli, E.; Ceppetelli, V.; Del Mastro, M.; Rotundo, L.; Brandi, G.; Magnani, M. Validation and application of a quantitative real-time PCR assay to detect common wheat adulteration of durum wheat for pasta production. Food Chem. 2017, 224, 86-91, https://doi.org/10.1016/j.foodchem.2016.12.053
- Schulze, C., Geuthner, AC. & Mäde, D. Development and validation of a method for quantification of common wheat, durum wheat, rye and barley by droplet digital PCR. Eur Food Res Technol. 2021, 247, 2267–2283, https://doi.org/10.1007/s00217-021-03786-y
- López, M.I.; Trullols, E.; Callao, M.P.; Ruisánchez, I. Multivariate screening in food adulteration: Untargeted versus targeted modelling. Food Chem. 2014, 147, 177-181, https://doi.org/10.1016/j.foodchem.2013.09.139.
- Miaw, C.S.W.; Martins, M.L.C.; Sena, M.M.; de Souza, S.V.C. Screening Method for the Detection of Other Allergenic Nuts in Cashew Nuts Using Chemometrics and a Portable Near-Infrared Spectrophotometer. Food Anal. Methods. 2022, 15, 1074-1084, doi:10.1007/s12161-021-02184-0.
- Zheng, L.; Bao, Q.; Weng, S.; Tao, J.; Zhang, D.; Huang, L.; Zhao, J. Determination of adulteration in wheat flour using multi-grained cascade forest-related models coupled with the fusion information of hyperspectral imaging. Spectrochim Acta A Mol Biomol Spectrosc. 2022, 270, 120813, https://doi.org/10.1016/j.saa.2021.120813.
- Zhao, X.; Wang, W.; Ni, X.; Chu, X.; Li, Y.-F.; Sun, C. Evaluation of Near-Infrared Hyperspectral Imaging for Detection of Peanut and Walnut Powders in Whole Wheat Flour. Appl. Sci. 2018, 8, 1076. https://doi.org/10.3390/app8071076
Reviewer 2 Report
Comments and Suggestions for Authors
I find the manuscript well written and organized.
2. Comprehensive Review of Major Plant Protein Sources
The title of table 1 should be above the table and not at the end.
2.6. Oat
Lines 146-147: the text says “It contains 12%–20% [52]”, I assume you are referring to protein content.
4. Detection Methods for Food Fraud in Plant-Based Proteins
4.1. Chromatography-Based Detection
Matrix effect, sample preparation and/or derivatization are not considered. I assume you are referring to protein content.
I would describe more strategies to detect potential allergens.
References
Line 562: the page is missing
Line 578: the page is missing
Line 585: the page is missing
Author Response
I find the manuscript well written and organized.
Comment 1) The title of table 1 should be above the table and not at the end.
Response to Comment 1) We thank the reviewer for the valuable comment. Following the suggestion, we have relocated the title “Table 1. Nutritional and functional properties of major plant proteins.” above the table, as indicated in Line 147.
Comment 2) Lines 146-147: the text says “It contains 12%–20% [52]”, I assume you are referring to protein content.
Response to Comment 2) We sincerely thank the reviewer for the insightful comment. The range “12%–20%” was indeed intended to refer to the protein content of oat. To clarify this point and enhance scientific accuracy, we have revised the sentence in Lines 218-219 to read: “The protein content of oat typically ranges from 12% to 20%.” We believe this revision improves the clarity and precision of the description.
Comment 3) 4.1. Chromatography-Based Detection. Matrix effect, sample preparation and/or derivatization are not considered. I assume you are referring to protein content. I would describe more strategies to detect potential allergens.
Response to Comment 3) We sincerely thank the reviewer for the insightful and valuable comment. We agree that matrix effects, sample preparation procedures, and derivatization strategies are critical factors influencing the reliability of chromatographic analyses. As this review aims to provide a broad overview of analytical methods and their reported performance in detecting food fraud across various plant-based protein targets, it was challenging to delve deeply into these technical aspects without disrupting the overall narrative structure.
Nevertheless, we fully acknowledge the importance of such methodological considerations. To address this point, we have added a brief discussion in Lines 364-370 outlining their potential impact on analytical performance, and we cite previous studies that provide comprehensive treatments of these topics.
We also concur with the need to more thoroughly address allergen detection. Accordingly, we have incorporated additional references to studies focusing on the detection of allergenic plant protein admixtures (Lines 372-376, 429-432, 470-473, and 493-495).
We appreciate the reviewer’s thoughtful suggestion, which has helped enhance the depth and comprehensiveness of our manuscript.
Lines 364-370) Chromatographic methods coupled with mass spectrometry demonstrate strengths in high sensitivity, specificity, and reproducibility, making them indispensable tools for food fraud detection. However, matrix effects, sample preparation procedures, and derivatization strategies significantly influence analytical accuracy and precision. Although beyond the scope of this overview, comprehensive discussions on these methodological considerations have been addressed extensively in previous studies (Williams et al., 2023).
Lines 372-376) Seki et al. developed an LC–MS/MS-based method capable of simultaneously detecting and quantifying trace contamination or admixture of allergenic grains, including wheat, spelt, rye, barley, oats, and buckwheat, in processed foods, with detection limits as low as 0.028–0.056 μg/L.
Lines 429-432) Schulze et al. developed a validated ddPCR assay that accurately quantified common wheat, durum wheat, rye, and barley, which are all recognized as allergenic cereals, and reliably detected their presence at levels as low as 1% in adulterated durum wheat.
Lines 470-473) Miaw et al. developed a portable NIR spectroscopy method that accurately detected cashew adulteration with allergenic nuts (peanut, Brazil nut, macadamia, pecan) at 0.1–10% w/w, enabling rapid, non-destructive allergen screening.
Lines 493-495) Zheng et al. demonstrated that HSI accurately detected and classified peanut and walnut adulterants in wheat flour at concentrations as low as 0.03% (w/w), highlighting its utility for non-destructive allergen screening in grain-based products.
References:
- Williams, M.L.; Olomukoro, A.A.; Emmons, R.V.; Godage, N.H.; Gionfriddo, E. Matrix effects demystified: Strategies for resolving challenges in analytical separations of complex samples. J Sep Sci 2023, 46, e2300571, https://doi:10.1002/jssc.202300571.
- Seki, Y.; Nakamura, K.; Arimoto, C.; Kikuchi, H.; Yamakawa, H.; Nagai, H.; Ito, T.; Akiyama, H. Development of a simple and reliable high-performance liquid chromatography–tandem mass spectrometry approach to simultaneously detect grains specified in food allergen labeling regulation on processed food commodities. J Chromatogr A. 2021, 1639, 461877, https://doi.org/10.1016/j.chroma.2021.461877.
- Schulze, C., Geuthner, AC. & Mäde, D. Development and validation of a method for quantification of common wheat, durum wheat, rye and barley by droplet digital PCR. Eur Food Res Technol. 2021, 247, 2267–2283, https://doi.org/10.1007/s00217-021-03786-y
- Miaw, C.S.W.; Martins, M.L.C.; Sena, M.M.; de Souza, S.V.C. Screening Method for the Detection of Other Allergenic Nuts in Cashew Nuts Using Chemometrics and a Portable Near-Infrared Spectrophotometer. Food Anal. Methods. 2022, 15, 1074-1084, doi:10.1007/s12161-021-02184-0.
- Zheng, L.; Bao, Q.; Weng, S.; Tao, J.; Zhang, D.; Huang, L.; Zhao, J. Determination of adulteration in wheat flour using multi-grained cascade forest-related models coupled with the fusion information of hyperspectral imaging. Spectrochim Acta A Mol Biomol Spectrosc. 2022, 270, 120813, https://doi.org/10.1016/j.saa.2021.120813.
Comment 4) References. Line 562: the page is missing. Line 578: the page is missing. Line 585: the page is missing
Response to Comment 4) We thank the reviewer for the careful and constructive comment. In response, we have added the missing page numbers to the corresponding references at Lines 705, 721, and 730 to ensure the accuracy and completeness of the citation information.
References:
- Paudel, D.; Dhungana, B.; Caffe, M.; Krishnan, P. A review of health-beneficial properties of oats. Foods. 2021, 10, 2591, https://doi.org/10.3390/foods10112591.
- Gonçalves, B.; Pinto, T.; Aires, A.; Morais, M.C.; Bacelar, E.; Anjos, R.; Ferreira-Cardoso, J.; Oliveira, I.; Vilela, A.; Cosme, F. Composition of nuts and their potential health benefits-an overview. Foods. 2023, 12, 942, https://doi.org/10.3390/foods12050942.
- Tian, L.; You, X.; Zhang, S.; Zhu, Z.; Yi, J.; Jin, G. Enhancing functional properties and protein structure of almond protein isolate using high-power ultrasound treatment. Molecules. 2024, 29, 3590, https://doi.org/10.3390/molecules29153590.
Reviewer 3 Report
Comments and Suggestions for Authors
April 21, 2025
Dear authors,
After reviewing the article titled “Food Fraud in Plant-Based Proteins: Analytical Strategies and 2 Regulatory Perspectives” which was submitted for possible publication in the Foods journal. I consider that the work addresses an interesting and ad hoc topic with the scope of the journal. However, I think that some aspects could help it improve.
Abstract
The authors must include methodological aspects of how the systematic review is carried out (search period, databases consulted, keywords, etc.).
Introduction
The introduction is well written and includes relevant information about the topic, the research problem, the background and concludes by mentioning the objectives of the research. I consider it to be adequate. I only have a suggestion:
- Some background on the study of Food Fraud in Plant-Based Proteins using the techniques compiled in Figure 1 should be mentioned.
It is imperative that authors include a section on the methodology applied to the review conducted, which should be separated from the results (sub-themes developed).
Summarize the search strategy, the selection of articles included in the review, the inclusion/exclusion criteria, the search period and the data extraction in one figure. I suggest checking:
- a) Kitchenham, B. (2004). Procedures for carrying out systematic reviews. Keele, UK, Keele University, 33(2004), 1-26.
- b) Arrizubieta, J. I., Ukar, O., Ostolaza, M., & Mugica, A. (2020). Study of the environmental implications of the use of metal powder in additive manufacturing and its handling. Metals, 10(2), 261. https://doi.org/10.3390/met10020261
- c) Ochoa-Noriega, C. A., Aznar-Sánchez, J. A., Velasco-Muñoz, J. F., & Álvarez-Bejar, A. (2020). The use of water in agriculture in Mexico and its sustainable management: A bibliometric review. Agronomy, 10(12), 1957. https://doi.org/10.3390/agronomy10121957
Line 95, Do not use abbreviations without first stating their meaning at least once. For example, GMO. Apply to the whole document.
Chickpea or chick pea? Standardize throughout the document
In tables 3 and 4 indicate the meaning of the abbreviations in the table footnotes.
Line 280, It is imperative that the detection limit is not expressed as a percentage; rather, the appropriate units should be indicated.
It is recommended that a section be added to the text on the extraction of protein from plant sources.
Conclusion and Future Prospects
The conclusions of the study are consistent with its objectives and are derived from the results. They are presented in a straightforward and accessible manner.
The reviewer considers that the manuscript may be suitable for publication following minor revisions.
King regards
Author Response
After reviewing the article titled “Food Fraud in Plant-Based Proteins: Analytical Strategies and Regulatory Perspectives” which was submitted for possible publication in the Foods journal. I consider that the work addresses an interesting and ad hoc topic with the scope of the journal. However, I think that some aspects could help it improve.
Comment 1) Abstract. The authors must include methodological aspects of how the systematic review is carried out (search period, databases consulted, keywords, etc.).
Response to Comment 1) We sincerely appreciate the reviewer’s insightful suggestion to include the methodological aspects of the systematic literature review in the abstract. In response, we have revised the abstract to clearly reflect the scope and rigor of our methodology. Specifically, we have added a sentence (Lines 20–23) that outlines the search period (2010–2025), the databases consulted (Scopus, Web of Science, and PubMed), and the use of structured keyword strategies related to plant-based proteins, food fraud, adulteration, and authentication. This addition is intended to improve methodological transparency and facilitate a clearer understanding of the systematic framework underpinning this review.
Lines 20–23) A systematic literature review was conducted using structured search strategies across Scopus, Web of Science, and PubMed, covering publications from 2010 to 2025 and incorporating keywords related to plant-based proteins, food fraud, adulteration, and authentication, thereby ensuring methodological rigor and comprehensive coverage.
Comment 2) Introduction. The introduction is well written and includes relevant information about the topic, the research problem, the background and concludes by mentioning the objectives of the research. I consider it to be adequate. I only have a suggestion: Some background on the study of Food Fraud in Plant-Based Proteins using the techniques compiled in Figure 1 should be mentioned.
Response to Comment 2) We sincerely thank the reviewer for the positive evaluation and thoughtful suggestion.
In response, we have revised the Introduction section to include a more specific and comprehensive explanation of Figure 1, and to clearly describe the analytical approaches commonly used for detecting food fraud in plant-based proteins. Specifically, we have added the following description (Lines 58–69) to contextualize the four major analytical categories—chromatography-based, DNA-based, spectroscopy-based, and imaging-based techniques.
We also noted that although these techniques are widely employed in general food fraud detection, they have been comparatively underutilized in the detection of adulteration and misrepresentation specifically within plant-based proteins. This was intended to underscore the need for increased research attention and the development of tailored analytical strategies in this field.
Furthermore, to address the reviewer’s suggestion for including background on food fraud in plant-based proteins, we have incorporated a representative example earlier in the Introduction (Lines 51–54). These illustrative cases highlight typical patterns of adulteration driven by economic incentives. A broader set of documented incidents and risk types is discussed in greater detail in Section 4 (Food Fraud Risks Associated with the Plant-Based Protein Industry).
We sincerely appreciate the reviewer’s helpful feedback, which contributed to improving the clarity, completeness, and scientific value of the manuscript.
Lines 51-54) For instance, chickpea flour, which commands a higher economic value due to its favorable nutritional profile, is sometimes adulterated with lower-cost alternatives like grass pea flour, and premium rice varieties such as basmati are occasionally mixed with common rice to artificially increase volume and profitability (Faller et al., 2021).
Lines 58-69) Analytical approaches employed in detecting food fraud, including fraud associated with plant-based proteins, encompass chromatography-based, DNA-based, spectroscopy-based, and imaging-based methods (Figure 1). Chromatography-based and DNA-based techniques are characterized by their high precision and accurate detection capabilities, making them particularly effective in laboratory environments. Conversely, spectroscopy-based and imaging-based methods offer advantages such as non-destructive analysis and high suitability for on-site applications. Each approach has distinct advantages depending on the specific context, and therefore, these methods have been extensively utilized in detecting food fraud (Vinothkanna et al., 2024). However, despite their effectiveness, these analytical techniques have primarily been applied to detecting fraud in animal-derived food products, and comprehensive studies specifically addressing food fraud related to plant-based protein sources remain limited.
References:
- Faller, A.C.; Kesanakurti, P.; Arunachalam, T. Chapter 14 - Fraud in grains and cereals. In Food Fraud, Hellberg, R.S., Everstine, K., Sklare, S.A., Eds.; Academic Press. 2021, pp. 281-308, https://doi.org/10.1016/B978-0-12-817242-1.00007-5.
- Vinothkanna, A.; Dar, O.I.; Liu, Z.; Jia, A.-Q. Advanced detection tools in food fraud: A systematic review for holistic and rational detection method based on research and patents. Food Chem. 2024, 446, 138893, https://doi.org/10.1016/j.foodchem.2024.138893.
Comment 3) It is imperative that authors include a section on the methodology applied to the review conducted, which should be separated from the results (sub-themes developed).
Summarize the search strategy, the selection of articles included in the review, the inclusion/exclusion criteria, the search period and the data extraction in one figure. I suggest checking:
- a) Kitchenham, B. (2004). Procedures for carrying out systematic reviews. Keele, UK, Keele University, 33(2004), 1-26.
- b) Arrizubieta, J. I., Ukar, O., Ostolaza, M., & Mugica, A. (2020). Study of the environmental implications of the use of metal powder in additive manufacturing and its handling. Metals, 10(2), 261. https://doi.org/10.3390/met10020261
- c) Ochoa-Noriega, C. A., Aznar-Sánchez, J. A., Velasco-Muñoz, J. F., & Álvarez-Bejar, A. (2020). The use of water in agriculture in Mexico and its sustainable management: A bibliometric review. Agronomy, 10(12), 1957. https://doi.org/10.3390/agronomy10121957
Response to Comment 3) We sincerely thank the reviewer for the valuable suggestion. The recommendation to clearly delineate the methodology applied to the systematic review, separate from the results section, has significantly contributed to enhancing the structural clarity and scientific rigor of the manuscript. This section explicitly describes the search strategy, search period (2010–2025), databases consulted (Scopus, Web of Science, and PubMed), keywords used in inclusion/exclusion criteria, and the multi-step screening and data extraction process.
Furthermore, we greatly appreciate the reviewer’s reference to key methodological sources, particularly Kitchenham (2004) and Arrizubieta et al. (2020). These references provided helpful guidance in structuring our methodological section and were instrumental in informing the development of Figure 2, which visually summarizes the SLR process in accordance with established best practices.
We believe these additions substantially improve the methodological transparency and reproducibility of the review.
Lines 84-113) 2. Methodological Framework for Literature Review
This review employs a systematic literature review (SLR) approach, drawing upon established methodological frameworks [Kitchenham (2024), Arrizubieta et al. (2020)], to comprehensively examine food fraud in plant-based proteins (Figure 2). To structure the investigation, two primary research questions were posed: (i) What are the common forms and documented instances of food fraud in plant-based protein products? and (ii) What analytical methodologies are utilized to detect adulteration and authenticate plant-derived protein sources? These questions in-formed both the search strategy and the inclusion/exclusion criteria. Accordingly, we constructed structured Boolean search queries combining the terms: ("plant-based protein" OR "plant protein" OR "alternative protein" OR "alternative food") AND ("food fraud" OR "adulteration" OR "authentication"). Searches were performed in Scopus, Web of Science (WoS), and PubMed, which were selected for their comprehensive disciplinary coverage and academic rigor, with the search period spanning from 2010 to 2025.
The identified articles underwent a rigorous multi-step screening and evaluation process. Initial quality verification ensured methodological rigor, scientific validity, and relevance to the review’s scope. Subsequent title screening ensured alignment with the re-search objectives, and clearly irrelevant studies were excluded. Articles meeting these criteria proceeded to full-text evaluation, where comprehensive assessment confirmed their applicability. Studies failing to meet the inclusion standards at any stage were excluded from the final selection.
In addition to the primary database search, a complementary "snowballing" approach was adopted [Wohlin (2014)], leveraging reference lists and citations from key articles identified during the primary search. This approach facilitated the identification of additional pertinent publications beyond the original search parameters, including studies published before 2010, provided they offered significant foundational or contextual value. The final selection comprises methodologically sound and contextually relevant publications that collectively enhance understanding and provide substantial insights into food fraud detection and authentication practices within the plant-based protein sector.
Figure 2. Systematic literature review workflow for identifying and selecting relevant studies
(Please see in manuscript)
References:
- Kitchenham, B. Procedures for Performing Systematic Reviews. Keele Univ. 2004, 33, 1–26.
- Arrizubieta, J.I.; Ukar, O.; Ostolaza, M.; Mugica, A. Study of the Environmental Implications of Using Metal Powder in Additive Manufacturing and Its Handling. Metals. 2020, 10, 261. https://doi.org/10.3390/met10020261
- Wohlin, C. Guidelines for snowballing in systematic literature studies and a replication in software engineering. In Proceedings of the 18th International Conference on Evaluation Assessment in Software Engineering, Berlin, Germany, 30 March 2014; p. 38.
Comment 4) Line 95, Do not use abbreviations without first stating their meaning at least once. For example, GMO. Apply to the whole document.
Response to Comment 4) We sincerely thank the reviewer for the insightful comment. As suggested, we have provided the full term “genetically modified organism (GMO)” upon its first appearance in the manuscript (Lines 166-169), and thereafter used the abbreviation “GMO” consistently throughout the text. Additionally, to maintain terminological consistency, all instances of “GM” have been standardized to “GMO” (Lines 166-169, 296-301, 333-334, 418-421).
Lines 166-169) In addition to its nutritional benefits, pea protein is highly valued for its cost-effective, genetically modified organism (GMO)-free, cholesterol-free qualities, as well as its relatively low allergenic potential when integrated into plant-based food formulations.
Lines 296-301) A notable example is the fraudulent labeling of GMO and non-GMO soybeans, which is driven by price disparities and consumer demand for non-GMO products. The higher market value of non-GMO soy creates economic incentives for fraudulent mislabeling, with instances of GMO soy being falsely marketed as non-GMO to exploit the premium status of non-GMO soy.
Lines 333-334) (Table 2.) Fraudulent labeling of GMO soy as non-GMO to exploit premium pricing.
Lines 418-421) By partitioning samples into thousands of micro-reactions, this method directly counts DNA targets and is thus highly effective in verifying non-GMO plant protein authenticity, for which precise quantification is key for regulatory compliance.
Comment 5) Chickpea or chick pea? Standardize throughout the document
Response to Comment 5) We thank the reviewer for this careful observation. We have standardized the term to “chickpea” throughout the manuscript, recognizing that this spelling is most widely adopted in major scientific literature and peer-reviewed journals. Accordingly, the sentence on Lines 198–200 has been revised to: “These nutritional and functional benefits establish the value of chickpea in the competitive alternative food industry.”
Comment 6) In tables 3 and 4 indicate the meaning of the abbreviations in the table footnotes.
Response to Comment 6) We sincerely thank the reviewer for this valuable suggestion. In accordance with the comment, we have added explanatory footnotes to clarify all abbreviations used in Table 3 (Lines 349-359) and Table 4 (Lines 532-537). This revision ensures greater clarity and accessibility of the tabular information for readers who may not be familiar with the technical terminology. We appreciate the reviewer’s attention to detail, which has contributed to improving the overall readability and consistency of the manuscript.
Lines 349-359) LC-MS/MS, liquid chromatography–tandem mass spectrometry; GC-MS, gas chromatography–mass spectrometry; HPLC, high-performance liquid chromatography; UPLC-MS/MS, ultra-performance liquid chromatography–tandem mass spectrometry; PCR, polymerase chain reaction; qPCR, quantitative polymerase chain reaction; ddPCR, droplet digital polymerase chain reaction; DNA, deoxyribonucleic acid; NGS, next-generation sequencing; FT-IR, Fourier-transform infrared spectroscopy; FT-MIR, Fourier-transform mid-infrared spectroscopy; NIR, near-infrared spectroscopy; FT-NIR, Fourier-transform near-infrared spectroscopy; HSI, hyperspectral imaging; SWIR-HSI, short-wave infrared hyperspectral imaging; VNIR-HSI, visible–near-infrared hyperspectral imaging; AI, artificial intelligence; GMO, genetically modified organism; LOQ, limit of quantification; LOD, limit of detection; R², coefficient of determination.
Lines 532-537) LC-MS/MS, liquid chromatography–tandem mass spectrometry; GC-MS, gas chromatography–mass spectrometry; HPLC, high-performance liquid chromatography; UPLC-MS/MS, ultra-performance liquid chromatography–tandem mass spectrometry; PCR, polymerase chain reaction; qPCR, quantitative polymerase chain reaction; ddPCR, droplet digital polymerase chain reaction; NGS, next-generation sequencing; FT-IR, Fourier-transform infrared spectroscopy; NIR, near-infrared spectroscopy; HSI, hyperspectral imaging; AI, artificial intelligence; GMO, genetically modified organism.
Comment 7) Line 280, It is imperative that the detection limit is not expressed as a percentage; rather, the appropriate units should be indicated.
Response to Comment 7) We sincerely thank the reviewer for the valuable and precise suggestion. In accordance with the comment, we have revised the expression of the detection limit to use an appropriate concentration unit rather than a percentage. Specifically, the sentence in Line 372 has been modified to state the detection limit as “<200 µg/g,” which corresponds to the originally indicated 0.02% (w/w). This revision improves clarity and ensures consistency with standard scientific reporting practices.
References:
- Spörl, J.; Speer, K.; Jira, W. A rapid LC-MS/MS multi-method for the detection of 23 foreign protein sources from legumes, oilseeds, grains, egg and milk in meat products. J Food Compost Anal. 2023, 124, 105628, https://doi.org/10.1016/j.jfca.2023.105628.
Comment 8) It is recommended that a section be added to the text on the extraction of protein from plant sources.
Response to Comment 8) We appreciate the reviewer’s thoughtful recommendation regarding the inclusion of a section on protein extraction from plant sources. We fully agree that this is a crucial aspect of understanding the properties and applications of plant-based proteins, and that its inclusion adds valuable context. However, because protein extraction methods vary considerably depending on the plant source and intended end-use, we were concerned that a detailed discussion of each method might detract from the main focus of this review—namely, the compositional and economic characteristics of plant proteins relevant to food fraud.
Therefore, rather than creating a standalone section, we have integrated a concise and broadly applicable overview of protein extraction strategies in Lines 126-146. This addition summarizes key approaches such as dry fractionation and wet extraction, including their principles, advantages, and limitations, and highlights how extraction choices affect protein purity and functionality. We believe this approach provides essential background without disrupting the narrative flow, while still acknowledging the importance of extraction methods as emphasized by the reviewer.
Lines 126-146) Protein extraction from plant sources is a critical step determining the nutritional, functional, and economic value of plant-based protein ingredients. Extraction methods can be broadly classified into dry fractionation and wet extraction processes, each with distinct advantages and limitations [Zhang et al., 2024]. Dry fractionation typically involves mechan-ical milling and air classification, utilizing differences in particle sizes and densities to enrich proteins without substantial water use, making it highly sustainable. However, this approach generally yields lower protein purity due to incomplete separation from other components [Meenakshi Sundaram et al., 2024]. In contrast, wet extraction methods, particularly alkaline solubilization coupled with isoelectric precipitation, are commonly used to achieve higher purity protein isolates (>95%) [Pulivarthi et al., 2023]. In these processes, proteins are initially solubilized at conditions away from their isoelectric points to remove insoluble impurities, and subsequently precipitated by adjusting pH near their isoelectric points, followed by separation and drying. Wet extraction methods often incorporate auxiliary technologies, including enzymatic hydrolysis, ultrasound, microwave, or high-pressure techniques, to enhance cell wall disruption and improve protein recovery rates. These modern technologies significantly increase ex-traction efficiency; however, they can incur higher operational costs and complexity. Therefore, the selection of extraction methods depends critically on the target plant source, intended product application, economic considerations, and desired functional properties of the final protein ingredient. These extraction strategies have been comprehensively re-viewed in previous studies, offering valuable insights into their optimization across di-verse plant protein sources and applications [Zhang et al., 2024, Hewage et al., 2022].
References:
- Zhang, X.; Zhang, T.; Zhao, Y.; Jiang, L.; Sui, X. Structural, extraction and safety aspects of novel alternative proteins from different sources. Food Chem. 2024, 436, 137712, https://doi.org/10.1016/j.foodchem.2023.137712.
- Meenakshi Sundaram, G.S.; Das, D.; Emiola-Sadiq, T.; Sajeeb Khan, A.; Zhang, L.; Meda, V. Developments in the Dry Fractionation of Plant Components: A Review. Separations. 2024, 11, 332. https://doi.org/10.3390/separations11120332.
- Pulivarthi, M.K.; Buenavista, R.M.; Bangar, S.P.; Li, Y.; Pordesimo, L.O.; Bean, S.R.; Siliveru, K. Dry Fractionation Process Operations in the Production of Protein Concentrates: A Review. Rev. Food Sci. Food Saf. 2023, 22, 4670–4697. https://doi.org/10.1111/1541-4337.13237
- Hewage, A.; Olatunde, O.O.; Nimalaratne, C.; Malalgoda, M.; Aluko, R.E.; Bandara, N. Novel Extraction technologies for developing plant protein ingredients with improved functionality. Trends Food Sci Technol. 2022, 129, 492-511, https://doi.org/10.1016/j.tifs.2022.10.016.